# Vaccination After Haematopoietic Stem Cell Transplant: A Review of the Literature and Proposed Vaccination Protocol

**DOI:** 10.3390/vaccines12121449

**Published:** 2024-12-23

**Authors:** André Silva-Pinto, Isabel Abreu, António Martins, Juliana Bastos, Joana Araújo, Ricardo Pinto

**Affiliations:** 1Infectious Diseases Department, São João Hospital, 4200-319 Porto, Portugal; isabel.abreu@ulssjoao.min-saude.pt (I.A.); antonio.pedro.martins@ulssjoao.min-saude.pt (A.M.); 2Faculty of Medicine, University of Porto, 4099-002 Porto, Portugal; mjuliana.bastos@ulssjoao.min-saude.pt (J.B.); joana.lima.araujo@ulssjoao.min-saude.pt (J.A.); rj.pinto@ulssjoao.min-saude.pt (R.P.); 3Haematology Department, São João Hospital, 4200-319 Porto, Portugal

**Keywords:** haematopoietic stem cell transplantation, immunization schedule, immunocompromised host

## Abstract

**Background/Objectives:** Haematopoietic stem cell transplantation (HCT) induces profound immunosuppression, significantly increasing susceptibility to severe infections. This review examines vaccinations’ necessity, timing, and efficacy post-HCT to reduce infection-related morbidity and mortality. It aims to provide a structured protocol aligned with international and national recommendations. **Methods:** A systematic review of current guidelines and studies was conducted to assess vaccination strategies in HCT recipients. The analysis included the timing of vaccine administration, factors influencing efficacy, and contraindications. Recommendations for pre- and post-transplant vaccination schedules were synthesised, specifically for graft-versus-host disease (GVHD), immunosuppressive therapy, and hypogammaglobulinemia. **Results:** Vaccination is essential as specific immunity is often lost after HCT. Inactivated vaccines are recommended to commence three months post-transplant, including influenza, COVID-19, and pneumococcal vaccines. Live attenuated vaccines remain contraindicated for at least two years post-transplant and in patients with ongoing GVHD or immunosuppressive therapy. Factors such as GVHD and immunosuppressive treatments significantly impact vaccine timing and efficacy. The review also underscores the importance of pre-transplant vaccinations and ensuring that patients’ close contacts are adequately immunised to reduce transmission risks. **Conclusions:** Implementing a structured vaccination protocol post-HCT is critical to improving patient outcomes. Timely and effective vaccination strategies can mitigate infection risks while addressing individual patient factors such as GVHD and immunosuppression. This review highlights the need for tailored vaccination approaches to optimize immune reconstitution in HCT recipients.

## 1. Introduction

Haematopoietic progenitor cell transplantation (HCT)—either allogeneic or autologous—induces a period of immunosuppression that increases the risk of severe and potentially life-threatening infections [1]. The development of transplant-related complications, such as the onset of graft-versus-host disease (GVHD) and its prophylaxis or the need for maintenance immunosuppressive therapies to prevent disease recurrence, may also contribute to sustaining this infectious risk [2].

However, some infections that can cause morbidity and mortality in this patient group are preventable through vaccination. Specific immunity to many infections is largely lost within weeks to months following transplantation, thus justifying the need for revaccination in this context [3].

Nevertheless, the decision to initiate vaccination in the post-transplant period must consider, on the one hand, the likelihood of achieving a minimally robust vaccine response in highly immunocompromised patients and, on the other hand, the risk of developing a potentially fatal infection post-transplant, which is particularly elevated during the first weeks to months after HCT.

Uncertainties regarding the timing of vaccination initiation, potential adverse side effects associated with vaccines, and the influence of certain factors (such as GVHD or the use of immunoglobulins) have led to unsatisfactory vaccination coverage in this patient group [4].

Therefore, this protocol aims to summarise the abovementioned points, considering the most recent international and national recommendations. A summary of the recommended inactivated and live attenuated vaccines post-HCT, along with their scheduling, is provided in the appendix of this document (Table 1, Table 2 and Table 3). When available, all recommendations are supported by cited bibliographic references. Where robust evidence is lacking, the guidance provided is based on expert opinion, reflecting current best practices and clinical experience.

Finally, the protocol offers some considerations regarding indications for pre-transplant vaccination and additional vaccination/precautions for close contact with transplanted patients.

## 2. Assessment of Conditions for Vaccination Following Haematopoietic Progenitor Cell Transplantation

All patients should undergo a clinical assessment three months post-transplant to initiate the vaccination programme. The considerations outlined in this section of the protocol take precedence over the therapeutic indications presented individually throughout the rest of the protocol.

### 2.1. Impact of GVHD and Immunosuppressive Therapy on Vaccination

For inactivated vaccines, delaying the start of vaccination in patients diagnosed with GVHD who receive a dose of prednisolone greater than 0.5 mg/kg/day (or equivalent) or three immunosuppressive drugs is recommended. Vaccines may be administered when the dose of prednisolone is lower than the aforementioned dose and/or one of the three concomitant immunosuppressive drugs is discontinued. In any case, vaccination initiation should not be postponed for more than three months [5].

It is important to note that the administration of live attenuated vaccines is generally contraindicated during the first two years post-transplant and in patients with GVHD or undergoing immunosuppressive therapy (including maintenance chemotherapy). In very specific cases, live vaccines may be administered after carefully assessing the patient’s epidemiological context and immunosuppression status [3,6,7].

### 2.2. Impact of Hypogammaglobulinemia on Vaccination

Patients with severe hypogammaglobulinemia (i.e., IgG levels below 3 g/L) are more susceptible to invasive bacterial infections caused by encapsulated bacteria such as Streptococcus pneumoniae or Haemophilus influenzae type B. Their response to vaccination is also diminished [8].

Vaccination should begin once the patient is no longer experiencing severe hypogammaglobulinemia. If recovery from hypogammaglobulinemia is not expected within three months, vaccination should commence, with the option to repeat doses if necessary.

### 2.3. Impact of Immunoglobulin Therapy on Vaccination

The administration of immunoglobulins (Ig) has minimal impact on the body’s response to inactivated vaccines. Therefore, the use of immunoglobulin-containing products does not preclude the administration of these vaccines, which can even be given concurrently [9].

On the other hand, the ability to respond to live attenuated vaccines may be adversely affected by immunoglobulin therapy. For this reason, it is recommended that live attenuated vaccines (MMR, varicella vaccine, or yellow fever vaccine) be administered only 8–11 months after the last administration of a full dose of immunoglobulins (corresponding to an Ig dose of 300–400 mg/kg) [9]. In specific situations of epidemic outbreaks, live attenuated vaccines may be administered three months after the infusion of immunoglobulins, although vaccine efficacy may be reduced [10,11]. Considering the epidemiological context, the decision to vaccinate in these cases should be made on a case-by-case basis.

### 2.4. Impact of Rituximab or Other Anti-CD20 Drugs on Vaccination

There is insufficient data to establish the maximum duration of immunosuppression induced by administering anti-CD20 drugs [12]. Therefore, we suggest that inactivated vaccines be administered six months after the last dose of the drug. In the case of live attenuated vaccines, these should be given 12 months after the last administration of the drug.

This recommendation represents a compromise between the expected immunosuppression of anti-CD20 therapies and the infection risk linked to delaying vaccination.

### 2.5. Impact of Donor Lymphocyte Infusions on Vaccination

The administration of donor lymphocytes post-allogeneic transplant is not expected to affect the response to inactivated vaccines negatively. Therefore, administering inactivated vaccines should be independent of donor lymphocyte infusion timing.

## 3. Inactivated Vaccines

The safety profile of inactivated vaccines in patients undergoing HCT appears to be like that of healthy individuals. Additionally, there is no evidence to date suggesting that this type of vaccine triggers or worsens GVHD.

Inactivated vaccines can be administered simultaneously with, before, or after other live or inactivated vaccines [9].

The description of each vaccine is detailed in the text below. A summary of the vaccination schedules is outlined in Table 1.

### 3.1. Influenza Vaccine (Influenza Virus)

**Background:** In immunocompromised individuals, influenza can progress to more severe conditions such as pneumonia, ARDS (acute respiratory distress syndrome), myocarditis, or Guillain–Barré syndrome. There is also a higher risk of concomitant infection or superinfection by bacterial or fungal agents. The incidence of secondary pneumonia due to influenza in patients undergoing HCT is about 33%, with associated mortality in these cases ranging from 15 to 28% [13]. The risk of progression to pneumonia seems particularly significant in patients with neutrophils < 500/μL, age ≥ 40 years, undergoing myeloablative conditioning regimens, and GVHD and recent (<30 days) corticosteroid use [14].

**Vaccination Start Post-Transplant and Schedule:** The influenza vaccination is recommended starting 6 months post-transplant. It should be administered annually at the beginning of autumn/winter (influenza season in the northern hemisphere). Vaccination should be maintained for at least 6 months after the patient discontinues all immunosuppressive therapy. For patients who maintain some degree/form of immunosuppression, vaccination can be administered annually for life.

**Booster/Booster Need and Schedule:** In patients with severe GVHD or severe lymphopenia (<500/μL or CD4 count <200/μL), a booster with a second dose of the vaccine 4 weeks after the first dose may be considered.

**Contraindications/Reasons for Postponement:** History of anaphylactic reaction to any component of the vaccine, particularly excipients or egg proteins; history of Guillain–Barré syndrome within 6 weeks of receiving a dose of the vaccine. In these cases, revaccination should be considered on a case-by-case basis, with postponement if there is a febrile syndrome or acute illness.

**Remarks:** In the event of a community outbreak or if the transplant is performed during the influenza season, the vaccine can be administered 3 months post-transplant. However, in these cases, a booster dose should be given 4 weeks after the first dose [6]. During the period when vaccination of the patient is not recommended, it is essential to reinforce the need to vaccinate close family members/household contacts. If the patient travels to other countries, the existence of seasonal period differences should be verified (in the southern hemisphere, the influenza season is between April and September), and consider that in certain tropical/subtropical countries, influenza occurs throughout the year.

### 3.2. COVID-19 Vaccine (SARS-CoV-2 Virus)

**Background:** Immunocompromised patients have a higher risk of developing severe COVID-19, including viral pneumonia and respiratory failure. In patients with hematologic malignancies, the mortality rate can range from 10 to 60% [15]. mRNA vaccines are considered the safest and most effective COVID-19 vaccines, particularly in immunocompromised patients. They also have the advantage of being easily updated to the circulating variant(s) of the SARS-CoV-2 virus that is most relevant at a given time [16]. In patients undergoing HCT, observational studies have shown that the serological response can reach up to 90% with a three-dose vaccination schedule, which performs better than the conventional two-dose regimen [17].

**Vaccination Start Post-Transplant and Schedule:** Vaccination is recommended to start 6 months post-transplant. The primary vaccination schedule includes three doses, with intervals between administrations of 4 weeks (6, 7, and 8 months post-HCT). The primary vaccination schedule should include the most updated version of the mRNA vaccine for the circulating variant(s).

**Booster/Booster Need and Schedule:** An annual booster dose of the mRNA vaccine should be administered according to the recommendations of the Directorate-General for Health. The booster dose should be administered 3 or more months after the primary vaccination schedule.

**Contraindications/Reasons for Postponement:** If there is a severe acute illness or fever, postpone until resolution. History of myocarditis/pericarditis within weeks of receiving a dose of the vaccine. In these cases, revaccination should be considered on a case-by-case basis. The highest incidence rate of myocarditis/pericarditis was observed in male adolescents and young adults after the second dose of the mRNA vaccine. No other specific contraindications apart from those mentioned in the introduction of this document.

**Remarks:** The COVID-19 vaccine can also be administered 3 months post-transplant if epidemiological circumstances justify it. Up to 5% of patients undergoing allogeneic HCT may develop cytopenia and GVHD after administration of the mRNA vaccine [18].

### 3.3. Pneumococcal Vaccine (Against Streptococcus pneumoniae)

**Background:** The risk of invasive pneumococcal disease is 3.8–5.0 cases per 1000 autologous transplants and 8.2–9.0 per 1000 allogeneic transplants [3]. The likelihood of a vaccine response is higher if the schedule starts with the conjugate pneumococcal vaccine. A booster dose 6 months after primary vaccination appears to enhance the immune response (booster effect), although there may be more local and systemic reactions to the vaccine [10,19]. There do not appear to be differences in the ability to generate a vaccine response if the schedule starts at 3 months post-HCT (versus ≥ 9 months) [20]. The new 20-valent conjugate pneumococcal vaccine offers the broadest serotype coverage among conjugate vaccines. Although there are no specific studies of this vaccine in HCT patients, a vaccination schedule that prioritises the 20-valent conjugate vaccine has a theoretical rationale and is recommended by the Advisory Committee on Immunization Practices (ACIP) of the Centres for Disease Control and Prevention (CDC) [21]. A multinational European study demonstrated that around 20% of *Streptococcus pneumoniae* carriers had serotypes included in the 23-valent polysaccharide vaccine (PPV23) but not in PCV13, and nearly 5% had serotypes included only in PPV23 and not in PCV20 [22]. Therefore, given the broader serotype coverage of the 23-valent polysaccharide vaccine, particularly in the current period of uncertainty, its administration remains an option to ensure extended serotype protection.

**Vaccination Start Post-Transplant and Schedule:** Vaccination is recommended to start 3 months post-transplant. All patients should start with the conjugate pneumococcal vaccine. The primary vaccination schedule includes three doses, with intervals between administrations of 2 months (3, 5, and 7 months post-HCT).

**Booster/Booster Need and Schedule:** A 20-valent conjugate pneumococcal vaccine booster dose should be administered 6 months after primary vaccination (13 months post-HCT). Although some international entities, in the case of the post-HCT vaccination schedule including four doses of the 20-valent conjugate vaccine, do not require the administration of the 23-valent polysaccharide pneumococcal vaccine [21], we consider that given the potential increase in serotype coverage (including 2, 17F, and 20), the 23-valent polysaccharide vaccine should be administered 2 months after the booster dose of the 20-valent conjugate vaccine. Revaccination against Streptococcus pneumoniae with the 23-valent polysaccharide pneumococcal vaccine can be performed 5 years after the last dose. This interval aims to maintain sustained protection over time.

The positioning of the recently FDA-approved 21-valent conjugate vaccine in the post-transplant vaccination schedule is not yet well-defined, as it does not include some important serotypes, such as 4 or 15B [23].

**Contraindications/Reasons for Postponement:** If there is a severe acute illness or fever, postpone until resolution. No other specific contraindications apart from those mentioned in the introduction of this document.

**Remarks:** A previous episode of invasive pneumococcal disease does not provide immunity against all serotypes and, therefore, does not change the indications or timing for vaccine administration.

### 3.4. Vaccine Against Haemophilus Influenzae Type B (HiB)

**Background:** Infection by *Haemophilus influenzae* type B can cause bacteraemia, pneumonia, and sinusitis in the immediate period after transplantation, particularly between 3 and 12 months. The vaccine response is high—80–95% after two to three doses—even when administered early after transplantation [3,5].

**Start of Vaccination after Transplantation and Schedule:** Vaccination should begin 6 months after HCT. The vaccination schedule consists of three doses, administered according to the 0, 2, 12 months schedule (6, 8, and 18 months after HCT).

**Need for booster and schedule:** No booster dose is needed.


**Contraindications/reasons for postponement:**


Individuals with a history of anaphylactic reaction to a previous dose.

In the presence of severe acute illness, with or without fever, wait until full recovery.

### 3.5. Vaccine Against Invasive Disease by Neisseria meningitidis

**Background:** Invasive meningococcal disease is a rare event, even in patients undergoing HCT. However, it has a high potential for morbidity and mortality and can occur even in the late post-transplant period (>100 days after HCT) [3]. In Europe, according to the latest data available from the ECDC, serogroup B was responsible for 64% of cases of invasive meningococcal disease with identified serogroup [24].

**Start of Vaccination after Transplantation and Schedule:** Vaccination can begin 6 months after transplantation with the conjugate meningococcal vaccine anti-ACYW135 (2 doses separated by 2 months) and anti-B (2 doses separated by 2 months).

**Need for booster and schedule:** No booster dose is needed.


**Contraindications/reasons for postponement:**


If there is severe acute illness or fever, postpone until resolution.

**Remarks:** There are no data on the use of the conjugate vaccine against serogroup B in adults over 50 years old.

### 3.6. Vaccine Against Diphtheria, Tetanus, and Pertussis (Tdap)

**Background:** Regarding diphtheria and tetanus, the loss of protective antibody titters during the first year post-allotransplant is described in more than half of the patients. In the case of pertussis, the risk of severe infection in patients undergoing HCT is unknown. Still, they are recognised as a risk group due to the loss of antibodies after transplantation [3]. The response capacity to vaccines seems to be high—85 to 100% after three doses in the case of the tetanus vaccine and 70–100% after three doses of the diphtheria vaccine—although it may be conditioned by the presence of GVHD and the conditioning regimens used for transplantation [3]. Vaccines with high doses of diphtheria toxoid (DTPa) are not usually recommended in healthy adults due to the higher risk of adverse effects associated with the vaccine. Although some transplant centres use it in adults to increase the likelihood of vaccine response [3], its use is not recommended [9].

**Start of Vaccination after Transplantation and Schedule:** Vaccination can begin 6 months after transplantation with Tdap. The recommended schedule with Tdap is three doses according to the following schedule: 0, 2, and 12 months (6, 8, and 18 months after HCT).

**Need for booster and schedule:** In adulthood, booster doses are given with the tetanus and diphtheria vaccine in reduced doses (Td) at the following ages: 25 years, 45 years, and 65 years. From the age of 65, a booster dose is recommended every 10 years.


**Contraindications/reasons for postponement:**


Individuals with a history of anaphylactic reaction to a previous dose.

In the presence of severe acute illness, with or without fever, wait until full recovery.

If Guillain–Barré syndrome or brachial neuritis occurs within 6 weeks after a previous dose of the tetanus vaccine or if an Arthus-type reaction occurs after the tetanus or diphtheria vaccine, the decision to continue the schedule must be made on a case-by-case basis.

If the patient has encephalopathy, has predisposing factors for seizures, or has neurological deterioration, it is suggested to wait until neurological stabilisation (precaution for the pertussis vaccine) [9].

### 3.7. Vaccine Against Poliomyelitis (IPV)

**Background:** Loss of immunity against poliomyelitis occurs early after transplantation, especially in the case of allogeneic transplantation. Additionally, the presence of GVHD accelerates this loss. Despite this, the vaccine response is high, around 80–100% after three doses of the inactivated vaccine. The probability of vaccine response is similar whether the schedule is started at 6 months post-HCT or at 18 months post-HCT [5,25].

**Start of Vaccination after Transplantation and Schedule:** Vaccination can begin 6 months after transplantation. Three doses should be administered at the following intervals: 0, 2, and 12 months.

**Contraindications/Reasons for Postponement:** The oral poliovirus vaccine (live attenuated) is absolutely contraindicated after transplantation due to the risk of inducing vaccine-associated poliomyelitis.

**Remarks:** In the case of vaccinating a baby with the oral poliovirus vaccine, the patient should not be in contact with the baby’s faeces for 4 weeks after vaccination (risk of inducing vaccine-associated poliomyelitis).

### 3.8. Vaccine Against Hepatitis B

**Background:** Immunization against hepatitis B virus, administered prior to transplantation, decreases steadily over time after transplantation, with a loss of protective titters described in up to 90% of patients 5 years post-transplant. Additionally, donor immunity against HBV does not seem to prevent this loss significantly [26]. The benefit of administering double doses of the vaccine at each administration is also unknown.

**Start of Vaccination after Transplantation and Schedule:** Vaccination is recommended to start 6 months after transplantation.

All patients should be evaluated serologically before starting vaccination with the following tests:•Hepatitis B surface antigen (HBsAg);•Antibody to hepatitis B core antigen (anti-HBc);•Antibody to hepatitis B surface antigen (anti-HBs).

Patients with prior exposure to hepatitis B (anti-HBc positive pre-HCT) should be under chemoprophylaxis with entecavir or tenofovir during the peritransplant period.

If all three markers are negative (HBsAg-, Anti-HBc-, Anti-HBs-), the patient should begin a vaccination schedule. A standard three-dose vaccination schedule is initiated, with doses administered as follows: the first dose at month 0 (beginning of vaccination), the second dose at month 1, and the third dose at month 6. One to two months after the third dose, anti-HBs antibody levels are reassessed to evaluate the effectiveness of the vaccination. If the anti-HBs titer is greater than 10 IU/L, the patient is considered to have achieved adequate immunity against hepatitis B, and no further action is required. If the anti-HBs titer is less than 10 IU/L, the patient has not achieved the desired immune response, and a new vaccination schedule is recommended using double doses to enhance the immune response.

Patients with positive anti-HBc and negative HBs antigen (regardless of anti-HBs status) should maintain hepatitis B reactivation prophylaxis (with entecavir or tenofovir) during the peri-transplant period. Six months after the transplant, they should begin hepatitis B vaccination with a 0-, 1-, and 6-month schedule. If, upon reassessment of the anti-HBs titer, the patient has a titer equal to or greater than 10 IU/L, chemoprophylaxis can be discontinued. However, chemoprophylaxis should be maintained if the titer remains below 10 IU/L.

**Need for Booster and Schedule:** None.

**Contraindications/Reasons for Postponement:** If there is a severe acute illness or fever, postpone until resolution. No other specific contraindications beyond those mentioned in the introduction of this document.

**Remarks:** If clinically indicated, the hepatitis B vaccine can be combined with the hepatitis A vaccine.

### 3.9. Vaccine Against Human Papillomavirus (HPV)

**Background:** Patients undergoing HCT appear to have a higher risk compared to the general population of HPV infection in the oral cavity, a higher prevalence of genital warts, and a higher risk of anal and cervical cancer [27].

Start of Vaccination after Transplantation and Schedule: Vaccination can begin 6 months after transplantation.

For adult individuals undergoing transplantation at age 45 or younger, HPV vaccination is recommended and should be performed with the 9-valent vaccine. The HPV vaccine has no proven efficacy in individuals over 45 years old and is, therefore, not recommended for this age group [28]. The vaccination schedule for adults includes three doses, according to the 0-, 2-, and 6-month schedule.

**Need for Booster and Schedule:** None.

**Contraindications/Reasons for Postponement:** Should not be administered to pregnant women. If there is a severe acute illness or fever, postpone until resolution. The presence of warts or a history of neoplasms associated with HPV does not contraindicate the vaccine.

### 3.10. Vaccine Against Varicella-Zoster Virus (Recombinant)

**Background:** The reactivation of varicella-zoster virus (herpes zoster) infection is more frequent in immunocompromised patients, and complications such as postherpetic neuralgia are occurring [29]. The inactivated recombinant herpes zoster vaccine is considered safe and effective in immunocompromised patients. In patients undergoing autologous HCT, the vaccine showed 68.2% efficacy in preventing herpes zoster and associated complications [30]. In the case of allogeneic HCT patients, only observational studies have demonstrated the vaccine’s safety, although they present heterogeneous results regarding its immunogenicity [31].

**Start of Vaccination after Transplantation and Schedule:** Vaccination is recommended to start 6 months after transplantation. The primary vaccination schedule includes two doses, with intervals between administrations of 2 months (6 and 8 months post-HCT).

**Need for Booster and Schedule:** None.

**Contraindications/Reasons for Postponement:** If there is a severe acute illness or fever, postpone until resolution.

**Remarks:** Serology is not recommended before vaccination. In the case of patients undergoing autologous HCT, vaccination can be considered 3 months after transplantation [32]. Antiviral prophylaxis should be continued for at least 1 month after complete vaccination, although vaccination does not confer any protection against infection by other herpes group viruses (including herpes simplex 1).

## 4. Live Attenuated Vaccines

The administration of live attenuated vaccines is completely contraindicated in the following situations:•Patients with GVHD;•Patients with suspected or evident relapse of the underlying disease;•Use of systemic immunosuppressive therapy in the last 6 to 12 months;•Patients with severe hypogammaglobulinemia (IgG < 3 g/L);•Recent immunoglobulin therapy in the last 8 to 11 months.

The description of each vaccine is explained in the text below. The summary of vaccination schedules is outlined in Table 2.

### 4.1. Measles, Mumps, and Rubella Vaccine (MMR)

**Background:** In patients with a high degree of immunosuppression, such as those undergoing HCT, measles can present with atypical symptoms/signs: some studies indicate that more than 50% of patients exhibit an atypical rash in terms of macroscopic characteristics and distribution pattern. Additionally, up to 20% of patients may not have a rash as part of the initial clinical presentation, which can delay diagnosis [33].

Measles can complicate into potentially fatal pneumonia and/or encephalitis. Neurological sequelae may remain after infection resolution.

The loss of immunity evaluated 5 years after allogeneic transplantation is 60% for measles, 73% for mumps, and 52% for rubella [3].

**Start of Vaccination after Transplantation and Schedule:** Vaccination can be considered 24 months after transplantation. A useful mnemonic for considering the start of the schedule is “2 1 8”, meaning more than 2 years after transplantation, more than 1 year after stopping systemic immunosuppression, and more than 8 months after the last administration of immunoglobulins [6].

The evaluation for vaccine indication should start with serology for measles (IgG), rubella (IgG), and mumps (IgG). The schedule includes two doses, with an interval of 2 months between the two vaccines.


**In Community Outbreaks:**


Patients may start the MMR vaccination schedule at 12 months post-transplant (but only if they are under low-grade immunosuppression). In case of exposure to possible, probable, or confirmed cases, the first dose should be administered within 72 h of exposure;

For patients contraindicated for the vaccine, passive immunization with immunoglobulin (IV IgG: 400 mg/kg single dose) is recommended, to be administered within the first 6 days after exposure.

**Need for Booster and Schedule:** None.

**Contraindications/Reasons for Postponement:** Severe acute illness, with or without fever: postpone until complete resolution. The use of immunoglobulin-containing products requires postponing the administration of MMR 8–11 months after the last administration of a full dose of immunoglobulins (corresponding to an Ig dose of 300–400 mg/kg) [9].

**Remarks:** Necessary precautions should be taken to avoid pregnancy within 4 weeks after vaccination. The vaccine can cause temporary anergy to the tuberculin test. If this test is needed, it is recommended to administer the tuberculin test on the same day as the MMR or 4 weeks after [10].

### 4.2. Live Attenuated Varicella-Zoster Virus Vaccine

**Background:** Primary infection with varicella-zoster virus is rare but potentially fatal due to the risk of disseminated infection with organ involvement, particularly pneumonia, encephalitis, or hepatitis [34]. Prophylactic use of acyclovir after transplantation (allogeneic or autologous) effectively prevents varicella-zoster virus infection, allowing the administration of a live attenuated vaccine to be postponed in the immediate post-transplant period.

**Start of Vaccination after Transplantation and Schedule:** The evaluation for the varicella vaccine should be performed 24 months after transplantation and after more than 1 year of stopping immunosuppression. The patient should not be on acyclovir. This evaluation should start with serology (IgG) for varicella. In seronegative patients, the vaccine can be considered.

The recommended schedule is two doses. The interval between doses should be 2 months.

**Need for Booster and Schedule:** None.

**Contraindications/Reasons for Postponement:** If severe acute illness or fever, postpone until resolution. No other specific contraindications.

**Remarks:** Patients should be informed that they may develop symptoms compatible with the disease within 3 weeks after vaccination. If symptoms appear, they should seek medical assistance and start antiviral treatment.

For patients needing the tuberculin test, it should ideally be performed from 6 weeks after vaccination, as the result may be compromised/unreliable due to the vaccine.

The vaccine should not be administered to the donor within 4 weeks before blood collection for transplantation.

If close contacts of the patient receive the varicella vaccine, contact with the patient should be avoided for 6 weeks after vaccination. Additionally, if a post-vaccination rash appears, contact with the patient should be postponed until the skin lesions are completely resolved.

**In Case of Close Contact with an Individual Diagnosed with Varicella or herpes zoster,** the following patients should receive specific varicella immunoglobulin:•Seronegative patients for VZV transplanted less than 24 months ago;•Seronegative patients for VZV transplanted more than 24 months ago but still under immunosuppression (e.g., due to chronic GVHD);•Patients in conditioning regimen for HCT with contact with individuals with post-vaccination rash;•Seropositive patients for VZV but immunocompromised due to high-dose systemic corticosteroid use or who received a T-cell-depleted graft and had contact with individuals with varicella, herpes-zoster, or post-vaccination rash.

Immunoglobulin should be administered within the first 10 days after contact. If immunoglobulin administration is not possible, consider administering valacyclovir (1 g orally every 8 h) for 22 days after exposure.

### 4.3. Live Attenuated Vaccines Not Recommended After Hematopoietic Stem Cell Transplantation

•BCG;•Live attenuated zoster vaccine;•Rotavirus vaccine;•Oral poliovirus vaccine (OPV).

## 5. Other Vaccines to Consider Individually

The following vaccines should be evaluated case-by-case, as they depend on individual risk factors or specific conditions, such as travel to tropical countries.

### 5.1. Respiratory Syncytial Virus Vaccine

Two new vaccines against respiratory syncytial virus (RSV) are available: monovalent adjuvanted and bivalent non-adjuvanted. Both have shown efficacy >80% in preventing lower respiratory tract infection by RSV in patients over 60 years old [35,36]. Although there are no specific studies of these vaccines in patients undergoing HCT, they may be considered in patients over 60 years old starting from 6 months after transplantation. If vaccination is decided, the adjuvanted vaccine may have an advantage due to its higher immunogenicity.

### 5.2. Yellow Fever Vaccine

The yellow fever vaccine is a live attenuated vaccine. Its administration is not without risk and can cause neurotropic or viscerotropic disease. This vaccine should only be considered from 24 months after transplantation (allogeneic or autologous). Until then, the patient should be strongly advised against travelling to yellow fever endemic areas, especially if there are disease outbreaks. Eligible patients for the vaccine are those who meet the criteria for live vaccines.

### 5.3. Hepatitis A Vaccine

Hepatitis A serology is recommended 6 months after transplantation. Individuals with negative serology may receive the vaccine (two doses separated by 6 to 12 months). Re-evaluation of the IgG antibody titer after vaccination should be considered to confirm the vaccine response. Without a response, the patient may be indicated for post-exposure prophylaxis with immunoglobulin. If there is a simultaneous indication for hepatitis B vaccination, the use of combined vaccines (HAV + HBV) is recommended.

### 5.4. Typhoid Fever, Cholera, Japanese Encephalitis, and Rabies Vaccines

All these vaccines are inactivated vaccines. In case of travel to risk areas for one or more of these diseases, the possibility of vaccination can be reviewed starting from 6 months after transplantation.

## 6. Pre-Transplant Vaccination

Although the effect of pre-transplant vaccination is presumably lost with hematopoietic stem cell transplantation, this vaccination may confer some protection for the peri-transplant period (during the conditioning regimen and immediate post-transplant period). Therefore, it is recommended that patients proposed for hematopoietic stem cell transplantation be vaccinated according to age and the national vaccination plan and, additionally, be vaccinated with the 20-valent pneumococcal conjugate vaccine, *Haemophilus influenzae* type b vaccine, meningococcal vaccine, and, during the autumn/winter period, influenza and COVID-19 vaccines [9]. Inactivated vaccines should be administered at least 2 weeks before the conditioning regimen, and live attenuated vaccines should not be administered 4 weeks before the collection of hematopoietic progenitor cells and the conditioning regimen.

## 7. Donor Vaccination Before Transplantation

Some vaccines have been described to elicit a superior response in the recipient in the post-transplant period when administered to the donor. However, this effect only applies to vaccines with a T-cell-dependent response and appears to be only transient. Therefore, due to ethical concerns and practicality, pre-transplant donor vaccination is not recommended beyond what is indicated according to age and the national vaccination plan [3]. Additionally, the 13-valent pneumococcal conjugate vaccine should be considered [9]. Inactivated vaccines should be administered more than 2 weeks before transplantation. Live attenuated vaccines are contraindicated in the donor within 4 weeks before transplantation due to the risk of transmission of vaccine strain-associated diseases.

## 8. Vaccination of Close Contacts/Family Members

The vaccination recommendations for close contacts of patients undergoing hematopoietic stem cell transplantation are as follows:•Review compliance with the national vaccination plan indicated for the age of family members.•Annual influenza and COVID-19 vaccine while the patient is considered immunocompromised.•Varicella and MMR vaccines for seronegative family members: For those receiving the varicella vaccine, contact with the patient should be avoided for 6 weeks after vaccination. For MMR, close contacts who have been recently vaccinated should avoid direct contact with the patient for 4 weeks after administration to minimise the risk of transmission.•If there are children in the household who have received the rotavirus vaccine, the patient should avoid contact with the child’s faeces (diaper handling; hygiene care) for 4 weeks after vaccination.

## 9. Conclusions

In conclusion, the post-hematopoietic stem cell transplant period presents unique challenges for infection prevention due to the profound immunosuppression experienced by patients. This review emphasises the critical role of timely and appropriate vaccination to restore immunity and mitigate the risks of severe infections (Table 3). The recommended protocol, which includes both inactivated and select live attenuated vaccines at specific intervals post-transplant, aims to balance the need for immunity with the patient’s ability to respond effectively to vaccines. Special considerations, such as the impact of graft-versus-host disease, hypogammaglobulinemia, and ongoing immunosuppressive therapies, are essential for tailoring the vaccination approach. The proposed vaccination schedule aims to improve transplant recipients’ outcomes and quality of life, aligning with the latest guidelines to provide comprehensive immunisation coverage in this vulnerable population.

## Figures and Tables

**Table 1 vaccines-12-01449-t001:** Inactivated vaccines to consider in adults undergoing allogeneic or autologous hematopoietic progenitor cell transplantation.

Vaccine	Interval After HCT to Start Vaccination (Months)	Recommended Doses and Interval Between Doses (Months)	Comments
Influenza vaccine	6	1 (annual)	May be started at 3 months during an outbreak, with a booster dose after 4 weeks. In cases of GVHD or severe lymphopenia (<500/μL or CD4 count <200/μL), consider 2 doses 4 weeks apart.
COVID-19 vaccine	6	3 (4 weeks between doses)	May be started at 3 months during an outbreak. Annual booster as recommended by health authorities.
20-valent pneumococcal conjugate vaccine	3	4 (3 initial doses; 1 booster) (2 months between doses; booster 6 months after initial doses, i.e., 13 months post-HCT)	
23-valent pneumococcal polysaccharide vaccine	15	1 (2 months after conjugate booster; 15 months post-HCT)	A booster dose may be given 5 years after the last 23-valent dose.
Haemophilus influenzae type B (HiB)	6	3 (0, 2, 12 months)	
Hepatitis B (HBV)	6	3 (0, 1, 6 months)	
Diphtheria, tetanus, and pertussis (Tdap)	6	3 (0, 2, 12 months)	
Inactivated polio (IPV)	6	3 (0, 2, 12 months)	
MenACWY	6	2 (2 months apart)	
MenB	6	2 (2 months apart)	Data on efficacy/safety unavailable for individuals over 50 years.
9-valent human papillomavirus vaccine	6	3 (0, 2, 6 months)	Up to 26 years (consider up to 45 years).
Recombinant zoster vaccine	6	2 (2 months apart)	

Note: An outbreak is defined as the occurrence of multiple cases of infection within a specific geographic area or population over a short period, exceeding the expected baseline incidence. Monitoring of outbreaks should be conducted by the relevant public health authority. In an outbreak, the vaccination protocol should be adapted accordingly.

**Table 2 vaccines-12-01449-t002:** Live attenuated vaccines to consider in adults undergoing allogeneic or autologous hematopoietic progenitor cell transplantation.

Vaccine	Interval After HCT to Start Vaccination (Months)	Recommended Doses and Interval Between Doses (Months)	Comments
Measles, mumps, rubella (MMR)	24	2 (0, 2 months)	May be given at 12 months post-HCT during an outbreak.
Varicella vaccine	24	2 (0, 2 months)	For patients on acyclovir, discontinue the day before vaccination and resume 14 days after.
Zoster vaccine	Contraindicated	-	-
BCG	Contraindicated	-	-
OPV	Contraindicated	-	-

Legend: BCG—Tuberculosis vaccine (Bacille Calmette-Guérin); OPV—oral polio vaccine.

**Table 3 vaccines-12-01449-t003:** Recommended vaccination schedule for patients undergoing hematopoietic progenitor cell transplantation.

		Months Post-Transplant											
Vaccines		3	4	5	6	7	8	12	13	15	18	24	26
Influenza ^1^	I	X											
SARS-CoV-2 ^1^	I	X	X	X									
Pneumo20	I	X		X		X		X					
Pneumo23	I									X ^2^			
HiB	I	X	X	X									
HBV	I	X	X	X									
Tdap	I	X	X	X									
IPV	I	X	X	X									
MenACWY	I	X	X										
MenB	I	X	X										
HPV ^3^	I	X	X	X									
Varicella-zoster	I	X	X										
MMR	V											X	X
Varicella-zoster	V											X ^4^	X ^4^

I—inactivated vaccine; V—live attenuated vaccine. ^1^—annual booster; ^2^—booster 5 years after the last Pneumo23 dose; ^3^—for individuals up to 26 years (consider up to 45 years); ^4^—consider for seronegative patients.

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
