# Peer review of "Vaccination After Haematopoietic Stem Cell Transplant: A Review of the Literature and Proposed Vaccination Protocol"

_vaccines, 2024, doi:10.3390/vaccines12121449_

Round 1

Reviewer 1 Report

Comments and Suggestions for Authors

This is an interesting overview of the current data in vaccination after hematopoietic stem cell transplantation, and also provides a clear guideline on vaccination schedules after transplantation.

I have some minor remarks:

- There's no difference in the schedule for auto and allo SCT recipients, and I cannot find a clear explanation why this decision was made, as the immune reconstitution is clearly different in both cases

- Frequently it is not clear on which data a certain guidance has been based: is it expert opinion, a study, ...?

- the advice to give the vaccines and Igs on different anatomical sites is not so clear: eg Igs are give IV or SC (belly or upper legs), while most vaccines are given IM (sometimes SC in case of bleedings issues) in the upper arms

- adverse effects of live vaccines in patients under Ig treatment should be specified: not only decreased response, but also risk of infection

- the advice to wait 8-11 months after the last dose of Igs is not classical: the reference seems to be a Spanish guideline: what is the rationale? Especially if you can also give it after 3 months. 

- please add whether we should measure B-cell count before vaccination and what the treshhold is to vaccinate, and why

- table 1: what is the definition of outbreak?

- table 1: is a bit confusing for the pneumococcal vaccination: so you give 3 initial vaccins at 3, 5 and 7 months, and then a 4rd (1st booster) at 13 months post allo-SCT? Usually only 3 are given, and then a booster with unconjugated vaccine at 12 months... And then you suggest to give unconjugated 23-valent vaccine at 12 months (which would be 1 month before the suggested conjugated booster), but in the table is stated 2 months after the conjugated, so at 15 months...

- please define severe lymphopenia, also treshhold for CD4 counts?(usuallu >200 is accepted for live vaccines)

- Pneumococcal vaccine: it would be very interesting to have information on the prevalnce and virulence of the 4 additional serotypes present in unconjugated 23-valent as compared to the conjugated 20-valent

- A previous episode of invasive pneumococcal disease does not confer immunity: on what is this based? - no reference is mentioned and this is a bit unexpected

- Remarks on HiB vaccination: presence in urine for 2 weeks: what lab results does this impact?

- If some recommendations are the same for (almost) every vaccine, maybe it would be better to make a separate part on general recommendations/contra-indications..., maybe add for example the impact of a severe acute illness or fever, to part 2 of the general introduction

- Vaccine against N. meningitidis: no date on the use of the conjugate vaccine against serogroup B --> so what do you propose?

- Vaccine against Tdap: how do you define encephalopathy? Also type in that sentence: if the patient has..., or has ...

- IPV: pts travelling to endemic areas: you already give a booster at 12 months after the first dose to everybody, so this advice is a bit strange. How long after that booster would you advice to give that additional booster?

- MMR: Remarks same as with IPV: if a baby is vaccinated, how long should the patient avoid contact? Both (and maybe also for other live vaccines) should maybe only be put in part 8, as it is now for Rotavirus vaccinee.

- Serology measurement: only not for VZV recommended? What about MMR?

- 8. Vaccination of close contacts / family members: guidance on COVID vaccination should be given!

- Table 3: numbering of foot notes is not correct: 2 times footnote 1 and no 3... Please correct

Author Response

We thank the reviewer for the thoughtful and detailed comments, which have significantly contributed to improving the clarity and quality of our manuscript. Below, we provide point-by-point responses to address each comment.

Comment 1: There's no difference in the schedule for auto and allo SCT recipients, and I cannot find a clear explanation why this decision was made, as the immune reconstitution is clearly different in both cases

Response 1: You are correct that immune reconstitution differs between auto and allo SCT recipients. However, there is also a loss of immune response with autologous transplantation, and currently, there is no robust evidence suggesting that vaccination schedules should differ in autotransplants or be guided by the loss of specific antibodies.

Comment 2: Frequently it is not clear on which data a certain guidance has been based: is it expert opinion, a study, ...?

Response 2: We acknowledge your concern regarding the clarity of data sources for the recommendations provided. Most of the guidance presented is evidence-based and appropriately referenced. However, in situations where robust data is lacking, the recommendations reflect expert opinion. To clarify this, we have added the following sentence to the introduction: 'When available, all recommendations are supported by cited bibliographic references. Where robust evidence is lacking, the guidance provided is based on expert opinion, reflecting current best practices and clinical experience.'"

Comment 3: the advice to give the vaccines and Igs on different anatomical sites is not so clear: eg Igs are give IV or SC (belly or upper legs), while most vaccines are given IM (sometimes SC in case of bleedings issues) in the upper arms

Response 3: The recommendation to administer vaccines and immunoglobulins at different anatomical sites originates from practices involving tetanus immunoglobulin, where this distinction has clinical relevance. For non-specific immunoglobulin administration, this precaution is less critical. Considering this, we have removed the recommendation from the text for clarity and to avoid confusion.

Comment 4:  adverse effects of live vaccines in patients under Ig treatment should be specified: not only decreased response, but also risk of infection

Response 4: The indications and contraindications for live vaccines are addressed in detail in Chapter 2.1 (Impact of GVHD and Immunosuppressive Therapy on Vaccination) and Chapter 4 (Live Attenuated Vaccines). In this section, we focus specifically on the Impact of Immunoglobulin Therapy on Vaccination, emphasising decreased vaccine response. The risk of infection is discussed in the aforementioned sections.

Comment 5: the advice to wait 8-11 months after the last dose of Igs is not classical: the reference seems to be a Spanish guideline: what is the rationale? Especially if you can also give it after 3 months. 

Response 5: This recommendation is based on Portuguese guidelines, where the waiting time depends on the immunoglobulin dose administered. For a full dose of immunoglobulins (corresponding to an Ig dose of 300–400 mg/kg), the recommended interval is 8–11 months. This aligns with the recommendations outlined in the article 'How I vaccinate blood and marrow transplant recipients' (Blood, 2016) and the Advisory Committee on Immunization Practices (ACIP): Use of Vaccines and Immune Globulins in Persons with Altered Immunocompetence.

Comment 6: please add whether we should measure B-cell count before vaccination and what the treshhold is to vaccinate, and why

Response 6: We appreciate your suggestion regarding B-cell count measurement. However, current guidelines do not universally recommend measuring B-cell counts before vaccination in hematopoietic stem cell transplant recipients. This is because vaccination timing is primarily determined by the post-transplant timeline, clinical stability, and immune reconstitution status (e.g., absence of severe lymphopenia or GVHD). While B-cell recovery is important for vaccine response, there is no established threshold to guide vaccination decisions.

Comment 7: table 1: what is the definition of outbreak?

Responde 7: An 'outbreak' is defined as the occurrence of multiple cases of infection within a specific geographic area or population over a short period, exceeding the expected baseline incidence. This typically refers to community transmission confirmed by public health authorities or an increase in cases within healthcare settings. Monitoring of outbreaks should be conducted by the relevant public health authority, and, in the event of an outbreak, the vaccination protocol should be adapted accordingly. A note was added to the table.

Comment8: table 1: is a bit confusing for the pneumococcal vaccination: so you give 3 initial vaccins at 3, 5 and 7 months, and then a 4rd (1st booster) at 13 months post allo-SCT? Usually only 3 are given, and then a booster with unconjugated vaccine at 12 months... And then you suggest to give unconjugated 23-valent vaccine at 12 months (which would be 1 month before the suggested conjugated booster), but in the table is stated 2 months after the conjugated, so at 15 months...

Response 8: The pneumococcal vaccination schedule begins before 6 months post-transplant, which can result in some loss of immunogenicity. For this reason, a booster dose of the conjugated pneumococcal vaccine is recommended 6 months after the primary series with conjugated vaccine. While the need for the polysaccharide vaccine is debatable—given that the 20-valent conjugated vaccine already offers broader serotype protection—we opted to include it due to the current period of uncertainty following the introduction of higher-valency conjugated vaccines. Therefore, the polysaccharide vaccine is scheduled 2 months after the fourth dose of the conjugated vaccine, at 15 months post-transplant. The table, however, was confusing and has been revised to improve clarity and comprehension.

Comment 9: please define severe lymphopenia, also treshhold for CD4 counts?(usuallu >200 is accepted for live vaccines)

Response 9: Severe lymphopenia is typically defined as an absolute lymphocyte count below 500 cells/μL. Regarding CD4 counts, we agree that a threshold of >200 cells/μL is widely accepted as a criterion for the safe administration of live vaccines. We will include this clarification in the text to ensure the guidance is explicit. 

Comment 10: Pneumococcal vaccine: it would be very interesting to have information on the prevalence and virulence of the 4 additional serotypes present in unconjugated 23-valent as compared to the conjugated 20-valent

Response 10: We appreciate your suggestion regarding the additional serotypes in the 23-valent polysaccharide vaccine (PPV23) compared to the 20-valent conjugated vaccine (PCV20). Polysaccharide vaccines, such as PPV23, offer broader protection by covering additional serotypes, and a multinational European study demonstrated that around 20% of Streptococcus pneumoniae carriers had serotypes included in PPV23 but not in PCV13, and nearly 5% had serotypes included only in PPV23 and not in PCV20. While specific data on the prevalence and virulence of the additional serotypes in immunocompromised patients remains limited, some of these serotypes are known to contribute to invasive pneumococcal disease. We will clarify this point in the text and emphasize the potential added value of the 23-valent vaccine, particularly in the context of broader serotype coverage and the ongoing uncertainties regarding the long-term impact of higher-valency conjugated vaccines.

Comment 11: A previous episode of invasive pneumococcal disease does not confer immunity: on what is this based? - no reference is mentioned and this is a bit unexpected

Response 11: We intend to convey that a previous episode of invasive pneumococcal disease (IPD) does not guarantee immunity against all Streptococcus pneumoniae serotypes, as immunity is largely serotype-specific. Therefore, having experienced IPD does not eliminate the need for vaccination. To address this, the sentence has been revised for improved clarity.

Comment 12: Remarks on HiB vaccination: presence in urine for 2 weeks: what lab results does this impact?

Response 12: The presence of HiB antigens in the urine can impact the results of urinary antigen tests for Haemophilus influenzae type B. However, since this test is not frequently used in clinical practice, we have removed the sentence for clarity.

Comment 13: If some recommendations are the same for (almost) every vaccine, maybe it would be better to make a separate part on general recommendations/contra-indications..., maybe add for example the impact of a severe acute illness or fever, to part 2 of the general introduction

Response 13: While we understand and appreciate your suggestion, and indeed considered structuring the document with a separate section for general recommendations and contraindications, we believe the current format serves an important purpose. This document is also intended to function as a reference for specific vaccines. Consolidating general recommendations could lead to the loss of key details when readers search for information on a particular vaccine. For this reason, we opted to include relevant recommendations and contraindications within each vaccine's section to ensure clarity and accessibility.

Comment 14: Vaccine against N. meningitidis: no date on the use of the conjugate vaccine against serogroup B --> so what do you propose?

Response 14: We acknowledge the limited data on the use of the conjugate vaccine against Neisseria meningitidis serogroup B in hematopoietic stem cell transplant (HCT) recipients. Despite this, given the high potential morbidity and mortality associated with invasive meningococcal disease, we propose administering the vaccine as per general recommendations for high-risk populations, starting 6 months post-transplant. This approach balances the known risks of disease with the absence of specific data for this patient group.

Comment 15: Vaccine against Tdap: how do you define encephalopathy? Also type in that sentence: if the patient has..., or has ...

Response 15: We define encephalopathy in this context as a condition characterized by altered mental status lasting more than 24 hours, including symptoms such as confusion, coma, or prolonged seizures, without an alternative identifiable cause.Additionally, we have corrected the typo in the sentence to read: 'if the patient has..., or has ...' to improve clarity.

Comment 16: IPV: pts travelling to endemic areas: you already give a booster at 12 months after the first dose to everybody, so this advice is a bit strange. How long after that booster would you advice to give that additional booster?

Responde 16: ou are correct, and we appreciate your observation. The advice regarding an additional booster for patients traveling to endemic areas has been removed to avoid redundancy, as a booster dose is already administered at 12 months post-vaccination.

Comment 17: MMR: Remarks same as with IPV: if a baby is vaccinated, how long should the patient avoid contact? Both (and maybe also for other live vaccines) should maybe only be put in part 8, as it is now for Rotavirus vaccinee.

Response 17: We appreciate your suggestion and agree that the information regarding avoiding contact after live vaccines, including MMR, should be more streamlined. To improve clarity, we have moved this recommendation to Part 8 (Vaccination of Close Contacts/Family Members), where similar guidance for the Rotavirus vaccine is already addressed. Specifically, patients should avoid contact with vaccinated infants for 4 weeks after administering live vaccines, such as MMR. 

Comment 18: Serology measurement: only not for VZV recommended? What about MMR?

Response 18: Thank you for your observation. We confirm that serology measurement is indeed recommended for MMR, as stated in the text: 'The evaluation for vaccine indication should start with serology for measles (IgG), rubella (IgG), and mumps (IgG).' This ensures that vaccination is appropriately targeted to seronegative individuals.

Comment 19: 8. Vaccination of close contacts / family members: guidance on COVID vaccination should be given!

Response 19: We appreciate your suggestion. Guidance on COVID-19 vaccination for close contacts and family members will be added to ensure comprehensive recommendations. 

Comment 20: Table 3: numbering of foot notes is not correct: 2 times footnote 1 and no 3... Please correct
Response 20: We have carefully reviewed Table 3, and the numbering of footnotes appears correct as presented. If there is any specific misunderstanding or discrepancy, we kindly ask for clarification to ensure consistency

Reviewer 2 Report

Comments and Suggestions for Authors

Dear authors

I hope this finds you all well. Regarding the review of manuscript number vaccines- 3352366, entitled "Vaccination After Haematopoietic Stem Cell Transplantation: A Review of the Literature and Proposed Vaccination Protocol". It is indeed an interesting and well-organised review, but some minor comments should be made.

1-    The addition of 1 or 2 figures will increase the suspense and break the boredom barrier of the reader.

2-    In line 291: Add Remarks.

3-    In line 344: Add Need for Booster and Schedule.

Author Response

We thank the reviewer for the constructive comments and suggestions. Below are our responses:

  1. The addition of 1 or 2 figures will increase the suspense and break the boredom barrier of the reader.
    Thank you for your suggestion. While we appreciate the value of figures in enhancing reader engagement, we believe that the current structure and content of the manuscript provide sufficient clarity without the need for additional figures.

  2. In line 291: Add Remarks.
    We have carefully reviewed the section on Tdap but do not have any additional remarks to include at this point, as the current information is already comprehensive and aligned with existing guidelines.

  3. In line 344: Add Need for Booster and Schedule.
    Thank you for your suggestion. We have added details regarding the need for booster doses and the recommended schedule in line 344 to ensure this information is clear and complete.